# Using the Soft Systems Methodology to Link Healthcare and Long-Term Care Delivery Systems: A Case Study of Community Policy Coordinator Activities in Japan

**DOI:** 10.3390/ijerph19148462

**Published:** 2022-07-11

**Authors:** Yuko Goto, Hisayuki Miura

**Affiliations:** Department of Home Care and Regional Liaison Promotion, National Center for Geriatrics and Gerontology, Obu 474-8511, Aichi, Japan; gotoyuko@ncgg.go.jp

**Keywords:** soft systems methodology, complex systems, medical care system, long-term care system, integrated community care system

## Abstract

Due to the rapidly aging population in Japan, the government has been attempting to link the healthcare delivery system with the long-term care delivery system. However, there are complex challenges that must be overcome to link the two systems. A new methodology should be used to organize complex community challenges and propose solutions. This study aimed to visualize the unique challenges and worldviews of interested parties in each community, using the soft systems methodology (SSM). We aimed to visualize issues and clarify challenges associated with linking the healthcare and long-term care delivery systems; in turn, clarifying the thought process behind solution proposals. We gathered information regarding those who are actively linking these systems in communities in a Japanese municipality (community care coordinators) and organized the information according to the SSM procedure. By organizing information using the SSM, we were able to summarize the present situations of the community healthcare and long-term care delivery systems, visualize issues, clarify challenges associated with linking these two systems, and propose solutions. The SSM may be useful in organizing complex community information and deriving solutions.

## 1. Introduction

Populations in developed countries are aging [1], and the number of patients with chronic illnesses is increasing worldwide [2]. In Japan, as the proportion of elderly people is reaching 30% [3], all municipalities, as special and administrative living units for residents, are required to link the healthcare delivery system and the long-term care delivery system. This is because of the increasing number of elderly patients who are, for example, repeatedly hospitalized with medical services due to chronic illness, and who then receive support that continuously uses a care-insurance-funded service after discharge. To develop a service that maintains involvement with each patient, it has been made mandatory to link the healthcare delivery system and the long-term care delivery system [4].

In Japan, there is a medical service system for the whole nation, and fees for service are paid for the medical services provided by healthcare professionals at hospitals and clinics. It is a medical care provision system that maintains “free access”, allowing patients to freely select any medical institution to receive medical care in Japan [5]. Under this system, many Japanese citizens do not regularly visit a specific primary care physician, but generally visit their preferred hospitals or clinics at their preferred time, which makes it difficult for patients to build long-term trust with a specific physician.

A long-term care insurance system [6,7] has been operational since the 2000 as a system with municipalities as the insurer, and all citizens aged ≥40 years must join this system. Long-term care insurance can be primarily used after the age of 65 years. There are facility-type services for long-term care facility residents, and home services for patients undergoing home care. In both services, qualified care managers make patient care plans. According to the plan, paid caregivers (helpers) and other paid professionals, such as social workers at care offices, provide nursing care, but family care is not paid. In-home care service staff meetings are held regularly to review the care plans with primary care physicians in charge of the clinics.

Under these complicated health insurance and long-term care insurance systems, many older residents that need support in making difficult decisions in Japan fail to receive sufficient support and are troubled because of it. For example, nurses and accounting officers have a different understanding of how patients undergo repeat hospitalizations due to chronic illness, as well as a different awareness of the challenges that patients experience. Furthermore, medical institutions have different levels of understanding, awareness, and responses to the situation. Thus, to link the healthcare delivery system and the long-term care delivery system, the understanding of situations and awareness of challenges should be more diverse as more people become involved. To promote such activities, extremely complex situations must be understood, challenges must be clarified, and solutions must be proposed.

In 2015, Japan mandated all municipalities to promote the linkage of the healthcare and long-term care delivery systems. For this purpose, coordinating staff, called community care coordinators, have been assigned to communities.

The activities of these healthcare coordinators are new. By understanding the complex situations of communities, summarizing challenges, and proposing solutions, the healthcare delivery system and the long-term care delivery system in communities can start to be linked together.

The soft systems methodology (SSM) is a methodology based on sot systems thinking, and was developed as an answer to the unsuccessful application of hard system methodology in “complex real-world situations” [8,9,10,11,12]. SSM aims to cope with those situations in which people in a problem perceive and interpret the world in their own way and make a decision about it, using standards and values that may not be shared by others [12]. SSM uses system ideas to formulate the following basic mental acts: perceiving, selecting, predicting, comparing, and deciding actions [9,10,11,12]. In SSM, each individual’s perception related to the field is presented first, and after different perceptions are aggregated by accommodation, several perceived problematic situations are selected. In turn, a conceptual action model is created against the situation. By comparing the model and current situation, a feasible and ideal action model is created. The SSM has not been employed to promote activities of community care coordinators so far. In the present study, we aimed to show the potential use of the SSM to understand complex situations and visualize challenges in communities to promote activities of community care coordinators, with the goal of linking the healthcare and long-term care delivery systems.

## 2. Materials and Methods

### 2.1. Design

This was a case study of one location, using an officially collected activity report on community care coordinators.

### 2.2. Data Collection

The authors (Y.G. and H.M.) have been progress administrators for the collaborative project in Aichi Prefecture since 2016, in which more than 20 community care coordinators of different municipalities participated in workshops. The community care coordinators were people in charge of promoting coordination between the two insurance-related medical systems. In Aichi Prefecture, they were mainly assigned to municipal offices and/or local medical associations. Workshops were held more than seven times per year. Group discussions were mainly held to identify issues and establish procedures to resolve them. We collected materials at every workshop and regular reports on structured questions about current issues and the progress of coordinating activities once a year from 2016–2019 before the COVID-19 pandemic. The materials at workshops included discussion content (paper) during group work. In this study of the 20 community care coordinators, we used the information from the three community care coordinators of one municipality because their municipality has an average population and medical system in Aichi Prefecture, and the challenges and current situations in promoting the cooperation of healthcare and long-term care were clear. The Aichi Prefecture allowed us to use the materials officially to enhance the quality of the workshop. We could not collect personal information, such as the gender and age of the coordinators.

### 2.3. Data Analysis

The present study aimed to achieve an understanding of complex situations and visualization of challenges in communities for healthcare coordinators to link the healthcare delivery system and the long-term care delivery system. Among the seven stages of SSM, we used up to the fourth stage to qualitatively analyze the data [12,13,14]. This time, it was grasped from the discussion materials and structured reports how each coordinator perceived the problem situation of the people needing care. Because the descriptions of the materials and reports could be grasped directly in meaningful sentences, text analyses, like content analysis, were not used. The Stages 1 and 2 of SSM shown below were summarized by the authors through the materials and the reports from the coordinators. The authors, in turn, constructed the conceptual model through the Stages 3 and 4 of SSM.

#### SSM

Stage 1. Problem situation considered (unstructured): An understanding of regional problems was sought from the viewpoints of the various people working in the region, using rich picture methodology.Stage 2. Problem situation expressed: Information was collected and a consensus regarding regional issues and problem-solving priorities was formed.Stage 3. Root definitions of relevant systems derived from rich picture analyses: One tool of this activity was “root definition”, which was a statement describing the activity system to be modeled. The formulation of a root definition was aided using the PQR formula, which answered the following questions: what should be done (P), how should it be done (Q), and why should it be done (R)? In contrast, learning and discussions were summarized by the CATWOE mnemonic: customer (C), actors (A), transformation (T), worldview (W), owner (O), and environmental constraints and enablers (E). To assess outcomes, indicators of efficacy, efficiency, and effectiveness were used.Stage 4. Conceptual models of systems described in the root definition: A conceptual model of one or more aspects of the problem situation was created, outlining a set of purposeful activities relevant to the situation.

### 2.4. Ethical Considerations

We complied with Japan’s research ethics protocol, Guidance on the Ethical Guidelines for Medical and Health Research Involving Human Subjects (revised 23 March 2021) [15,16]. The data used in this study were previously reported; thus, the study was not subject to the research ethical board related to medical research involving human subjects [15,16].

## 3. Results

### 3.1. Stage 1. Problem Situation Considered (Unstructured)

Since the national policy to promote the linkage of the healthcare and long-term care delivery system came out in 2015, healthcare coordinators contacted professionals in the healthcare delivery system (e.g., physicians and nurses at hospitals, medical social workers, family doctors) and professionals in the long-term care delivery system (e.g., care managers, social workers at care offices, paid caregivers (helpers), visiting nurses, and pharmacists in communities). They individually visited these professionals and collected information on challenges associated with linking the two systems.

From this interview information, the coordinators perceived the past situation as follows: (1) for the past 6 years, there has been a forum for those involved in each system to gather in an informal manner, gather information from each other’s perspective, and exchange opinions regarding their own work. Those who participated in this space clarified that there are more people who believe that it is necessary to informally link the healthcare and long-term care delivery systems in the community. (2) In addition, it was acknowledged that it is a problem that informal activities cannot change community systems. (3) There was also a clear desire for formal activities to link the healthcare delivery system and the long-term care delivery system by a formal organization.

### 3.2. Stage 2. Problem Situation Expressed

Based on the information obtained in Stage 1, mayors of municipalities have begun taking actions to link both systems as a formal activity. They have also gathered representatives of professions related to healthcare systems, including physicians at local hospitals and family doctors, and those of long-term care system, including care managers and visiting nurses in a formal setting. The municipality managed these meetings, including rewards for attending professionals. The coordinators in the municipality coordinated these meetings, based on the results of Stage 1. These meetings were held multiple times over 6 months to visualize community issues while building consensus.

Issues presented by professionals related to the healthcare delivery system were visualized as follows upon consensus:Elderly people receiving care while being treated for an illness tended to have initial symptoms of exacerbation overlooked, leading to an increase in emergency admissions once their condition becomes serious (no primary physician system, but free access in Japan).The number of patients needing urgent care is increasing, and hospitals are constantly running out of beds (patients’ dependence on hospital care).An increasing number of elderly patients are readmitted within 1 month of discharge, leading to an increased number of patients requiring urgent care, and to bed shortages (shortage of transitional care).

Issues presented by professionals related to the long-term care delivery system were visualized as follows upon consensus:Even if there are concerns about patients (i.e., decreased appetite) while providing care, the staff cannot consult family doctors and visiting nurses because they are unfamiliar with the medical jargon (basic shortage of communication between healthcare and long-term care professionals).The staff would like advice from hospital doctors and nurses regarding diet and medications before patients are discharged, but there is no opportunity for such consultation (shortage of cooperation system between hospital care and community care after discharge).When patients have an issue, all the staff can do is call an ambulance (many care staff from welfare jobs).We summarized these issues as follows. Since there is no information exchange, advice, and consulting framework or relationship between those involved with the long-term care delivery system and with the health care delivery system, the burden is put on medical institutions as a locus for providing adequate management of illnesses for elderly patients (Figure 1).

### 3.3. Stage 3. Root Definitions of Relevant Systems Derived from Rich Picture Analyses

We performed an XYZ analysis to clarify the direction for issues in Stage 2 (Table 1).

We further clarified the results of the XYZ analysis with a CATWOE analysis (Table 2).

### 3.4. Stage 4. Conceptual Model of Systems Described in the Root Definition

We prepared a conceptual model based on the XYZ and CATWOE analyses of Stage 3. In this conceptual model, we placed the Z of the XYZ analysis (i.e., “create a system to provide healthcare and long-term care in a community where elderly patients’ conditions can be safely managed in their daily lives”) at the top and placed the X at the bottom. In this manner, we systematically analyzed the process of moving from Z to X, which is the goal (Figure 2).

## 4. Discussion

In the present study, we aimed to clarify the actions of healthcare coordinators, in the hopes of understanding complex community situations and visualizing challenges using the SSM as the thought process. The end goal was to link the healthcare and long-term care delivery systems in communities.

As seen in the rich picture (Figure 1), this is a complicated human activity involving the adjustment of the different perceptions and intentions of the parties concerned. In our case, the SSM was used as a method to visualize this activity. In Stages 1 and 2, the issues the coordinators perceived were highlighted. That is, in the healthcare system, the issues were no primary physician system, patients dependent on hospitals, insufficient transitional care system, and no continuity of care.

The use of ambulances and hospital emergency departments by older adults with mild illnesses is common throughout Japan. For context, it is important to mention that Japan does not have a system for allocating patients to clinics. There is no primary physician system, unlike in many countries in Europe and the United States. Japan has a free access system that allows patients to freely visit hospitals and community clinics. Though this free access guarantees patients’ freedom of choice, there is no official group of physicians that act as gatekeepers who can refer patients to hospitals when necessary. This situation has been stressing emergency staff in Japan, and this has been further exacerbated by the coronavirus disease 2020 pandemic. The lack of primary physicians as gatekeepers can also affect readmission immediately after discharge. Thus, we were challenged in using SSM to clarify the role of coordinators and, in turn, clarify their required activities under a system without primary physicians.

The issues of long-term care clarified in Stages 1 and 2 were all due to a shortage of communication between healthcare and long-term care professionals. One of the reasons is that many Japanese care staff are from welfare jobs who lack medical knowledge.

As a coping method for such a situation, the authors constructed a conceptual model for the solution using SSM (Figure 2, Stages 3 and 4). By the conceptual model, the direction of the actual action was clarified. The final goal was “the action creates rules in which those involved in the healthcare and the long-term care system can routinely exchange important information regarding the changes elderly patients experience during their routine care”. Without the SSM method, it would have been difficult to achieve such a clear direction. By feeding this result back to the coordinator, it is possible to make a more realistic action plan after comparing it with the current situation.

Healthcare and long-term care have developed under different insurance and education systems in Japan. As a result, different philosophies, cultures, and values have been formed in each field. In this situation, it was suggested that SSM is useful as a guide for visualizing differences and problems in a single community, integrating the current situation in complex areas, and building a community-based integrated care system, as conducted in a previous report [17]. SSM is a qualitative research methodology that identifies individual problems and directs the improvements of those situations. Because it elevates the ability to improve situations through systems thinking, it has been used for developing human resources and organizations in various fields [18]. Furthermore, by repeating the SSM learning process in a cyclic manner, the solutions are expected to be sustainable [19]. Previous studies have used SSM for improving healthcare policy, quality, and learning processes in the field of professional education [10,18,19,20]. In this study, we used SSM Stages 1–4 to identify complex situations and visualize challenges in communities to promote the activities of community care coordinators, with the goal of linking the healthcare and long-term care delivery systems. As with our case, the SSM could be applied to various aspects of the Japanese system. This can allow us to clarify the process of extracting problems and improving the quality of community care, as well as ensure that residents continue receiving integrated medical and long-term care in their communities.

To fulfill their missions, coordinators need a direction for their issue-oriented problem resolution. They need to act toward action goals in an issue-oriented manner rather than to perform uniform tasks, as described in Table 1. This study showed that SSM could be a powerful tool for achieving this aim. In order for coordinators to provide efficient leadership, operational research knowledge tools such as the SSM can be used.

### Strengths and Limitations

In this study, SSM analysis was performed with care coordinators from Aichi Prefecture. However, there are more than 1700 large and small municipalities in Japan, some of which do not have hospitals. Furthermore, their medical and long-term care resources differ, along with residents’ attitude toward medical treatment. Thus, this analysis for Aichi Prefecture may not be applicable to other Japanese municipalities. Because unpublished information was not included, there might be a bias in the data used in this study.

## 5. Conclusions

To support community activities that link the healthcare delivery system and the long-term care delivery system in Japan, the SSM could be useful in summarizing complex information of communities, and visualizing challenges based on the awareness of community healthcare coordinators.

## Figures and Tables

**Figure 1 ijerph-19-08462-f001:**
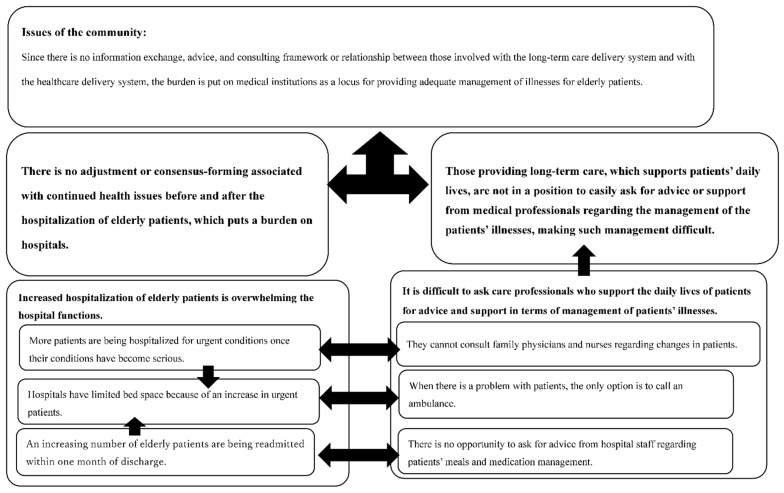
Visualization of information on community issues (rich picture).

**Figure 2 ijerph-19-08462-f002:**
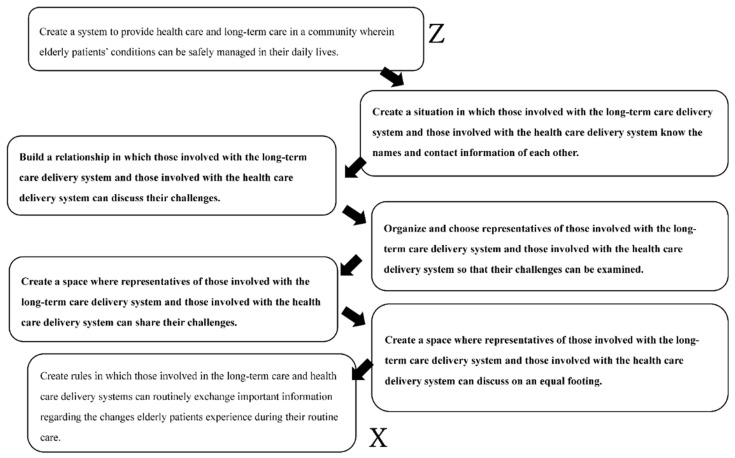
Conceptual model.

**Table 1 ijerph-19-08462-t001:** Results of the XYZ analysis.

X	Create rules in which those involved in the long-term care and healthcare delivery systems can routinely exchange important information on changes elderly patients’ experience within their routine care support. If there are any issues regarding the management of meals and medications during the hospitalization of patients, there should be an opportunity to advise those involved with the long-term care delivery system at the hospital before patients are discharged. Create rules for those in the long-term care delivery system to provide a space to discuss responses to patients whose conditions change quickly.
Y	Create a framework in which those involved with the healthcare delivery system and the long-term care delivery system can share information, advise, and consult/reduce the readmission rate within a month after discharge.
Z	Create a system to provide healthcare and long-term care in a community wherein elderly patients’ conditions can be safely managed in their daily lives.

**Table 2 ijerph-19-08462-t002:** CATWOE analytical results.

(C) customer	Elderly patients who receive healthcare while also receiving long-term care from those involved with the long-term care delivery system.
(A) actor	Those involved with the long-term care delivery system and with the healthcare delivery system.
(T) transformation	Admitted after conditions become serious →Prevent worsening of conditions during daily care →Reduced emergency readmission rate and readmission rate within 1 month of discharge
(W) worldview	A community wherein there is a connection between essential workers involved in patients’ conditions so that they can exchange information, give advice, and consult as necessary.
(O) owner	Mayor of municipality
(E) environmental constraints and enablers	The laws, insurance fees, payments to professionals, providing organizations, and consultation services for residents are different between the two systems. These aspects should be integrated quickly.

## Data Availability

Not applicable.

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
