# Peer review of "Using the Soft Systems Methodology to Link Healthcare and Long-Term Care Delivery Systems: A Case Study of Community Policy Coordinator Activities in Japan"

_ijerph, 2022, doi:10.3390/ijerph19148462_

Round 1

Reviewer 1 Report

I was really excited to read this paper as Integrating health and community services is an important, yet difficult issue worldwide. So you have an important topic, now you need to organize this so that readers can understand the methodology, the context, and then how your findings relate to the context.

The introduction leads to the need for the research. After that this paper becomes difficult for readers to follow.

I would suggest that after your introduction, before Materials and Methods you describe the SSM methodology and then the Japanese healthcare system and long-term care system.  Who lives in the long-term care system? Who provides care? Are families involved in care?  Where does primary care fit?  Are people living in community homes part of the long-term care system?

Material and methods

You need to organize this section. Don’t repeat yourself. You tell us several times that this was a report on one setting but not in a way that your reader can understand why that setting was selected or how many care coordinators were involved. You tell us twice that there were 20 community care coordinators and then that you used the information from one organization. We these from one municipality? One municipal office? one medical association? Why was that organization selected over others? Can you describe your population sample? How many of the 20 coordinators participated? Men, Women, Ages, Occupations?

Data collection—

Rewrite the first sentence: authors is plural and “has been a progress administrator” is singular. Are you talking about 2 progress administrators? What is the Aichi Prefecture? This needs to be explained in the context section.

There were meetings—did you collect data at the meetings? If so how? Did you record the meetings and transcribe the data? You want to describe your data collection methods in enough detail that another researcher could read this paper and be able to replicate your data collection and analysis methods.

Data Analysis 

Here you repeat yourself “Among the seven stages of SSM, 78 we used up to the fourth stage to qualitatively analyze the official data published by 79 healthcare coordinators [9–11].” and then  “In this study, we used SSM stages 1–4 to understand complex situations and visualize challenges in communities to promote activities of community care coordinators, with the goal of linking the healthcare and long-term care delivery systems.”

How did you qualitatively analyze the data? Did you use thematic analysis[1-3]? Content analysis[4,5]? grounded theory[6,7]? Interpretive description[8-11].  There are a number of qualitative data analysis methods and all have prescribed methods. I suggest reading the references above.  

Results:

Without understanding how you collected the data and then the methods by which you analyzed the data it is difficult to understand how you got the results. In each section, you need to order your results so that your reader understands the sequence or the list.  See Owl Purdue article on ordering. https://owl.purdue.edu/owl/graduate_writing/graduate_writing_topics/graduate_writing_organization_structure_new.html

In Stage 1—this might be ordered by time—e.g. For the first 6 years there was informal gathering of data, There were two problems with this:1) participants wanted more formal linkage and 2) participants acknowledged that informal activities could not change community systems. Then how was the problem defined? And by whom?

Stage 2 there is who and how this was organized and then  are lists from two perspectives

All of your results must be organized and described so the reader without knowledge of the Japanese context can understand them.

Discussion

Once you have rewritten the methods and results, I think your discussion will likely need to be tweaked slightly. You introduce primary care physicians in the discussion and I don’t seem to see any reference to that in the results.  

  I would suggest that this paper needs to be significantly revised  with a much better description of the context, methods and participants. 

1.            Braun, V.; Clarke, V. Using thematic analysis in psychology. Qualitative Research in Psychology 2006, 3, 77-101.

2.            Braun, V.; Clarke, V. Conceptual and Design Thinking for Thematic Analysis. Qualitative Psychology 2021, 9, 3-26, doi:10.1037/qup0000196.

3.            Braun, V.; Clarke, V. Can I use TA? Should I use TA? Should I not use TA? Comparing reflexive thematic analysis and other pattern-based qualitative analytic approaches. Counselling and Psychotherapy Research 2021, 21, 37-47, doi:10.1002/capr.12360.

4.            Brazil, K.; Howell, D.; Marshall, D.; Critchley, P.; Van Den Elzen, P.; Thomson, C. Building primary care capacity in palliative care: Proceedings of an interprofessional workshop. Journal of Palliative Care 2007, 23, 113-116.

5.            Hsieh, H.F.; Shannon, S.E. Three approaches to qualitative content analysis. Qualitative Health Research 2005, 15, 1277-1288, doi:10.1177/1049732305276687.

6.            Charmaz, K. Grounded Theory: Objectivist and Constructivist Methods. Handbook of Qualitative Research B2 - Handbook of Qualitative Research 2000, 509-535.

7.            Charmaz, K. Shifting the grounds: Constructivist grounded theory methods. Developing Grounded Theory: The Second Generation 2009, 127-154.

8.            Attridge, M.; Creamer, J.; Ramsden, M.; Cannings-John Hawthorne R, K. Culturally Appropriate Health Education for People in Ethnic Minority Groups with Type 2 Diabetes Mellitus 2014.

9.            Thorne, S. The science and art of theoretical location. Evidence-Based Nursing 2014, 17, 31, doi:10.1136/eb-2014-101738.

10.          Thorne, S. Interpretive description; Left Coast Press: Walnut Creek, CA, 2008.

11.          Thorne, S.; Jensen, L.; Kearney, M.H.; Noblit, G.; Sandelowski, M. Qualitative metasynthesis: Reflections on methodological orientation and ideological agenda. Qualitative Health Research 2004, 14, 1342-1365, doi:10.1177/1049732304269888.

Author Response

1. COMMENT: I was really excited to read this paper as Integrating health and community services is an important, yet difficult issue worldwide. So you have an important topic, now you need to organize this so that readers can understand the methodology, the context, and then how your findings relate to the context.

RESPONSE: Thank you for our comment. We have rewritten this manuscript to understand methodology, the context, and then how our findings relate to context better.

2. COMMENT: The introduction leads to the need for the research. After that this paper becomes difficult for readers to follow.

RESPONSE: We have rewritten this manuscript, according to your suggestion.

3.COMMENT: I would suggest that after your introduction, before Materials and Methods you describe the SSM methodology and then the Japanese healthcare system and long-term care system. Who lives in the long-term care system? Who provides care? Are families involved in care? Where does primary care fit?  Are people living in community homes part of the long-term care system?

RESPONSE: Thank you for your comments. We have added several sentences to clarify the SSM methodology, healthcare system, and long-term care system in Japan (Lines 35-50, 69-79).

Material and methods

4. COMMENT: You need to organize this section. Don’t repeat yourself. You tell us several times that this was a report on one setting but not in a way that your reader can understand why that setting was selected or how many care coordinators were involved. You tell us twice that there were 20 community care coordinators and then that you used the information from one organization. We these from one municipality? One municipal office? one medical association? Why was that organization selected over others? Can you describe your population sample? How many of the 20 coordinators participated? Men, Women, Ages, Occupations?

RESPONSE: Thank you for your comments. We have added the sentences about data collection (Lines 90-98) to clarify these points.

Data collection

5. COMMENT: Rewrite the first sentence: authors is plural and “has been a progress administrator” is singular. Are you talking about 2 progress administrators? What is the Aichi Prefecture? This needs to be explained in the context section.

RESPONSE: We have rewritten the first sentence (Line 84).

6. COMMENT: There were meetings—did you collect data at the meetings? If so how? Did you record the meetings and transcribe the data? You want to describe your data collection methods in enough detail that another researcher could read this paper and be able to replicate your data collection and analysis methods.

RESPONSE: Thank you for your comment. We have added the sentences to clarify this point (Line 103-107).

Data Analysis

7. COMMENT: Here you repeat yourself “Among the seven stages of SSM, 78 we used up to the fourth stage to qualitatively analyze the official data published by 79 healthcare coordinators [9–11].” and then “In this study, we used SSM stages 1–4 to understand complex situations and visualize challenges in communities to promote activities of community care coordinators, with the goal of linking the healthcare and long-term care delivery systems.”

RESPONSE: We have deleted the second sentences (Lines 131-133).

8. COMMENT: Line 90-98

There are some recommendations or guidelines to translate questionnaires in health care research. Which one did you depend on? If not any, why?

Is there a reasonable explanation why you did not do a pilot study, which is usually recommended to perform when translating health care related scales?

RESPONSE: We have added the sentence to clarify the point (Line 135-136).

The materials we analyzed were previously reported ones. We could not do a pilot study.

9. COMMENT: How did you qualitatively analyze the data? Did you use thematic analysis[1-3]? Content analysis[4,5]? grounded theory[6,7]? Interpretive description[8-11]. There are a number of qualitative data analysis methods and all have prescribed methods. I suggest reading the references above.

RESPONSE: Thank you for your polite advice and information provision. Because the descriptions of the materials and the reports could be grasped directly in meaningful sentences, text analysis like content analysis was not used. We have added the explanation about it (Lines 105-107).

Results:

10. COMMENT: Without understanding how you collected the data and then the methods by which you analyzed the data it is difficult to understand how you got the results. In each section, you need to order your results so that your reader understands the sequence or the list. See Owl Purdue article on ordering. https://owl.purdue.edu/owl/graduate_writing/graduate_writing_topics/graduate_writing_organization_structure_new.html

RESPONSE: We thank you for your information including Owl Purdue article. We confirmed it, and have added several sentences about how we get our results (Lines 103-109)..

11. COMMENT: In Stage 1—this might be ordered by time—e.g. For the first 6 years there was informal gathering of data, There were two problems with this:1) participants wanted more formal linkage and 2) participants acknowledged that informal activities could not change community systems. Then how was the problem defined? And by whom?

RESPONSE: We have corrected the sentences in Stage 1, and clarified who perceived the problem (Lines 143-151).

12. COMMENT: Stage 2 there is who and how this was organized and then are lists from two perspectives

RESPONSE: We have added the sentences to clarify this point (Lines 161-164).

13. COMMENT: All of your results must be organized and described so the reader without knowledge of the Japanese context can understand them.

RESPONSE: We have added several information related to Japanese system (Lines 170-188).  

Discussion

14. COMMENT: Once you have rewritten the methods and results, I think your discussion will likely need to be tweaked slightly. You introduce primary care physicians in the discussion and I don’t seem to see any reference to that in the results.

RESPONSE: We have rewritten the discussion drastically, and referred to primary physician system (Lines 211-239).

15. COMMENT: I would suggest that this paper needs to be significantly revised with a much better description of the context, methods and participants.

RESPONSE: We have corrected the text throughout the manuscript, according to your advice.

Reviewer 2 Report

1.      Because the data were extracted from the official published documents, the authors need to describe how many workshops, group discussions, and the other related documents were used in the section of materials and methods.

2.      Although the authors describe the seven stages of SSM, and used up to the fourth stage to qualitatively analyze the official data, the extraction process or data analysis was not clear.

3.      There are gaps between the results and methods due to the problems mentioned above.

4.      Since SSM is a qualitative research methodology, the authors should describe the process of data analysis.

Author Response

1. COMMENT: Because the data were extracted from the official published documents, the authors need to describe how many workshops, group discussions, and the other related documents were used in the section of materials and methods.

RESPONSE: Thank you for your comment. We have added several sentences in Methods to clarify the point (Lines 81-98).

2. COMMENT: Although the authors describe the seven stages of SSM, and used up to the fourth stage to qualitatively analyze the official data, the extraction process or data analysis was not clear.

RESPONSE: We have added several sentences in Methods to clarify the point (Lines 103-109).

3. COMMENT: There are gaps between the results and methods due to the problems mentioned above.

RESPONSE: We challenged to fill the gap by adding explanation about Japanese healthcare and long-term care system (Lines 161-188).

4. COMMENT: Since SSM is a qualitative research methodology, the authors should describe the process of data analysis.

RESPONSE: We have added several sentences in introduction (Lines 65-74) and method (Lines 103-109) to clarify the method of SSM.

Round 2

Reviewer 1 Report

Generally, this paper has been improved. The story of your research is quite clear. I really enjoyed the discussion.  Now the paper needs to be edited to improve its readability. Check the verb tense throughout the paper. For the most part, you could be using past tense. I have made some suggestions below and will attach the PDF with highlights and notes.

Introduction

Again what I am looking for in the introduction is an understandable explanation of the problem that explains the need for the research.  You have done this.

Small point here, Is “rewards” the usual way that you refer to “fee for service” or payment for medical services?  “In Japan, there is a medical service system for the whole nation, and rewards are paid for medical services provided by healthcare professionals at hospitals and clinics.

In the sentences, lines 40-43,

 “A long-term care insurance system [6,7] has been operational since the 2000 as a system with municipalities as the insurer, and all citizens aged ≥40 years must join this system. Long-term care insurance can be primarily used after the age of 65 years; municipalities manage this system.”

You have municipalities as the insurer and the manager of the system in 2 different sentences. It seems like a bit of a tack-on or afterthought in the second sentence.

2.2. Data Collection 83

“The authors (YG and HM) has have been a progress administrator for the collaborative project 84 in Aichi Prefecture since 2016, in which more than 20 community care coordinators of different 85 municipalities participated in workshops.”

Authors is plural—thus it has to read “The authors have been progress administrators

Data analysis

Lines 103-105

This time, it was grasped from the discussion materials and structured reports how each coordinator perceived

the problem situation of the people involved.

Perhaps something like this might be clearer, The discussion materials and structured reports described how each coordinator perceived the problem situation of the people needing care.

2.3.1. SSM 110

                      • Stage 1. Problem situation considered (unstructured): An understanding of regional problems 111 is sought from the viewpoints of the various people working in the region, using rich picture 112 methodology. 113

                      • Stage 2. Problem situation expressed: Information is collected and a consensus regarding re-114 gional issues and problem-solving priorities is formed. 115

                      • Stage 3. Root definitions of relevant systems derived from rich picture analyses: One tool of 116 this activity is “root definition,” which is a statement describing the activity system to be 117 modeled. The formulation of a root definition can be aided using the PQR formula, which 118 answers the following questions: what should be done (P), how should it be done (Q), and 119 why should it be done (R)? In contrast, learning and discussions may be summarized by the 120 CATWOE mnemonic: customer (C), actors (A), transformation (T), worldview (W), owner 121 (O), and environmental constraints and enablers (E). To assess outcomes, indicators of effi-122 cacy, efficiency, and effectiveness are used. 123

                      • Stage 4. Conceptual models of systems described in the root definition: A conceptual model 124 of one or more aspects of the problem situation is created, outlining a set of purposeful activ-125 ities relevant to the situation. 126

                      • Stage 5. Comparison of conceptual models with the real world situation: Regional stakehold-127 ers conduct investigations and organize to increase the feasibility of their initiatives. 128

                      • Stage 6. Identification of changes that are systematically desirable and culturally feasible: A 129 concrete action plan is established. 130

                      • Stage 7. Action to improve the problem situation: Identification of opportunities for improve-131 ment based on previous activities.

This list describes what your data analysis and the SSM methodology very nicely.  In academic writing you are going to use the past tense

Lines 150 From these the interview information, the coordinators perceived the past situation as follows:1) for the past 6 years, there has been a forum for those involved in each system to gather in an informal manner, gather information from each other’s perspective, and exchange opinions regarding their own work. Those who participated in this space clarified that there are were more people who believed that it is necessary to informally linking the healthcare and long-term care delivery systems in the community.; 2) in addition, it was acknowledged that it is a problem thatwith  informal activities is that they cannot change community systems. 3) There was also a clear desire for formal activities to link the healthcare delivery system and the long-term care delivery system by a formal organization.

Stage 2 Lines 166 to 169  These first 2 sentences say almost the same thing.

The challenges associated with linking the healthcare delivery system and the long-term

care delivery system in communities are summarized as follows:

Issues presented by professionals related to the healthcare delivery system were visualized as

follows upon consensus:

Then you repeat the second sentence again in line 178

Issues presented by professionals related to the long-term care delivery system were visualized as follows upon consensus:

Discussion

Lines 216- 217 This is an incomplete sentence, “That is, in the healthcare system, no primary physician system, patients’ dependent on hospitals, insufficient transitional care system.” You need a verb and perhaps to say the problem is no continuity of care.

Author Response

1.COMMENT: Generally, this paper has been improved. The story of your research is quite clear. I really enjoyed the discussion. Now the paper needs to be edited to improve its readability. Check the verb tense throughout the paper. For the most part, you could be using past tense. I have made some suggestions below and will attach the PDF with highlights and notes.

RESPONSE: Thank you for your comments. We have corrected the sentences, according your comments throughout the manuscript.

Introduction

2.COMMENT: Again what I am looking for in the introduction is an understandable explanation of the problem that explains the need for the research. You have done this.

Small point here, Is “rewards” the usual way that you refer to “fee for service” or payment for medical services?  “In Japan, there is a medical service system for the whole nation, and rewards are paid for medical services provided by healthcare professionals at hospitals and clinics.

RESPONSE: We have the word ‘rewards’ to ‘fee for service’ (line 34).

In the sentences, lines 40-43,

  1. COMMENT: “A long-term care insurance system [6,7] has been operational since the 2000 as a system with municipalities as the insurer, and all citizens aged ≥40 years must join this system. Long-term care insurance can be primarily used after the age of 65 years; municipalities manage this system.”

You have municipalities as the insurer and the manager of the system in 2 different sentences. It seems like a bit of a tack-on or afterthought in the second sentence.

RESPONSE: We have deleted the second sentence (line 43).

2.2. Data Collection

  1. COMMENT: “The authors (YG and HM) has have been a progress administrator for the collaborative project in Aichi Prefecture since 2016, in which more than 20 community care coordinators of different municipalities participated in workshops.” .

Authors is plural—thus it has to read “The authors have been progress administrators”

RESPONSE: We have corrected the sentence (line 84).

Data analysis

  1. COMMENT: Lines 103-105

This time, it was grasped from the discussion materials and structured reports how each coordinator perceived the problem situation of the people involved.

Perhaps something like this might be clearer, the discussion materials and structured reports described how each coordinator perceived the problem situation of the people needing care.

RESPONSE: We have corrected the sentence (lines 104-105).

  1. COMMENT:

2.3.1. SSM 110

  • Stage 1. Problem situation considered (unstructured): An understanding of regional problems is sought from the viewpoints of the various people working in the region, using rich picture methodology.
  • Stage 2. Problem situation expressed: Information is collected and a consensus regarding regional issues and problem-solving priorities is formed.
  • Stage 3. Root definitions of relevant systems derived from rich picture analyses: One tool of this activity is “root definition,” which is a statement describing the activity system to be modeled. The formulation of a root definition can be aided using the PQR formula, which answers the following questions: what should be done (P), how should it be done (Q), and why should it be done (R)? In contrast, learning and discussions may be summarized by the CATWOE mnemonic: customer (C), actors (A), transformation (T), worldview (W), owner (O), and environmental constraints and enablers (E). To assess outcomes, indicators of efficacy, efficiency, and effectiveness are used.
  • Stage 4. Conceptual models of systems described in the root definition: A conceptual model of one or more aspects of the problem situation is created, outlining a set of purposeful activities relevant to the situation.
  • Stage 5. Comparison of conceptual models with the real world situation: Regional stakeholders conduct investigations and organize to increase the feasibility of their initiatives.
  • Stage 6. Identification of changes that are systematically desirable and culturally feasible: A concrete action plan is established.
  • Stage 7. Action to improve the problem situation: Identification of opportunities for improvement based on previous activities.

This list describes what your data analysis and the SSM methodology very nicely.  In academic writing you are going to use the past tense

RESPONSE: We have corrected these sentences to use the past tense (lines 110-125) and deleted lines 126-131, because we did not use those in this study.

  1. COMMENT: Lines 150 From these the interview information, the coordinators perceived the past situation as follows:1) for the past 6 years, there has been a forum for those involved in each system to gather in an informal manner, gather information from each other’s perspective, and exchange opinions regarding their own work. Those who participated in this space clarified that there are were more people who believed that it is necessary to informally linking the healthcare and long-term care delivery systems in the community.; 2) in addition, it was acknowledged that it is a problem that with informal activities is that they cannot change community systems. 3) There was also a clear desire for formal activities to link the healthcare delivery system and the long-term care delivery system by a formal organization.

RESPONSE: We have corrected the sentence (line 152).

  1. COMMENT: Stage 2 Lines 166 to 169

These first 2 sentences say almost the same thing.

The challenges associated with linking the healthcare delivery system and the long-term care delivery system in communities are summarized as follows:

Issues presented by professionals related to the healthcare delivery system were visualized as follows upon consensus:

Then you repeat the second sentence again in line 178

Issues presented by professionals related to the long-term care delivery system were visualized as follows upon consensus:

RESPONSE: We have deleted the sentence to avoid the repetition (lines 162-163).

Discussion

  1. COMMENT: Lines 216- 217 This is an incomplete sentence, “That is, in the healthcare system, no primary physician system, patients’ dependent on hospitals, insufficient transitional care system.” You need a verb and perhaps to say the problem is no continuity of care.

RESPONSE: We have corrected the sentence (lines 210-211).

Reviewer 2 Report

The authors have revised the manuscript according to the comments suggested by the reviewer,

Author Response

1.COMMENT: The authors have revised the manuscript according to the comments suggested by the reviewer.

RESPONSE: Thank you for your comments. We believe this manuscript was marked improved by reviewers’ advice.
